# Picturing, Pledges and Other Scripted Acts: Performance (In) Art

## Dave Beech

School of Design, University of the Arts London, London WC1V 7EY, UK; d.beech@chelsea.arts.ac.uk

**Abstract:** This essay re-examines several examples of non-performance based artworks from the perspective and history of performance art. For instance, the photomontages I produced as an undergrad art student were based on repeated acts of stealing posters from their public sites on the street between my house and the college. Later, while working collaboratively with Mark Hutchinson in Manchester and London, we made photograms and paintings of scenes that we enacted on the street (striking a match, taking a flag for a walk). Years later, as a solo artist again, I produced a body of work in the form of slide presentations that consisted of photographs of me dressed in homemade costumes that cast me as a monster. In my work with the Freee art collective, we used slogans, our own bodies, costumes and props to give material reality to slogans that we treated as scripts. We wrote manifestos and staged events in which participants read. So, while the forms of my work are image based, the rationale is typically tied to the techniques and values of performance.

**Keywords:** picturing; script; pledge; walk; March; slogan; manifesto

## 1. Introduction

I have no idea what happened to all the drawings and paintings, photos and objects I made as an art student in the 1980s. I am not much of an archivist of my own work. My practice has been characterised more by self-critique than self-care. Everything has been lost except one slide. The earliest of my own artworks that I have a record of is a montage consisting of one poster cut into another. The image is striking but unremarkable. Visually, it is parasitic on a combination of the aesthetics of slick marketing and the early twentieth century avant-garde technique of repurposing items of popular culture for the gallery. Nevertheless, I still like the unserious and un-self-conscious montage mainly because I remember the feeling I had when I stole the two posters from outside a newsagent.

I noticed the poster but carried on walking down the street. I stopped and decided to take it out of the metal lattice frame that held it in place. I walked back and had a quick look at the way the frame was fixed to the wooden surface. No security, just a clip. In a split second I opened it up and grabbed the image, accidentally also taking the poster underneath it. Then I ran.

I have been involved in a handful of performance works but recalling these would hardly scratch the surface of the significance of performance within my practice. It is the performative elements of my non-performance works that establish the strongest connection between my practice and performance art. Performance seeps into contemporary art without causing a commotion. Like "Text Art", which no longer exists despite the fact that more artists use text in their works today than ever did during the heyday of Text Art, performance is present in contemporary art without separating itself from painting, sculpture, photography and installation art.

## 2. Picturing Is a Verb

When I worked collaboratively with Mark Hutchinson in the late-1980s and early-1990s, we made images out of the agenda set by Terry Atkinson and Art & Language. We

were particularly interested in Terry's use of sump oil and other materials that resulted in stuff falling off the paintings and landing on the floor of the gallery, especially his argument that these fragments of the painting suddenly become the responsibility of the curator, not the artist. Mike and Mel made some paintings by mouth, presented a group of paintings in boxes so they could not be seen, squashed paintings against sheets of glass so that the pictures were lost and glued posters almost as large as the paintings directly onto the surfaces of the paintings—not only treating paintings as objects (as sculpture?), but also inserting them into a sequence of events.

We made "invisible drawings" by dragging pencils across very large sheets of paper that, we imagined, left some kind of trace but not enough for the human eye to detect. These took hours and hours to produce, both of us stood side by side making scratching noises, in a process that had to be carefully monitored so that we did not go over the same area twice by mistake. After days or weeks, there was nothing to see except perhaps a few mistakes where one of us had pressed too hard and actually left a mark or even a fragment of a drawing visible on the paper. The activity of drawing was followed through in a strict, disciplined manner with a result, however, that needed an explanation. In other words, what we had we had *done* in making the work was not legible in the work itself. An invisible drawing, therefore, is the performance of a drawing that results in a minimally modified material object but no image.

The performance of the drawings was, it seemed to us, the primary feature of the work, but it was possible for this to be overlooked in the face of an art object that, despite appearing to be empty, complied with the conventions of art objects intended primarily to be judged on what stands before the observer. We wanted to thematise and dramatise the actions that we were taking but were not interested in the kind of heroic acts or bodily displays that we saw in the processes of Yves Klein, Jackson Pollock, Joseph Beuys and the like. Our solution was to make a small series of large photograms that equated lighting a match with changing the world. Thinking about Marx's famous thesis on Feuerbach that philosophers had only interpreted the works whereas the point is to change it, we looked for a real but minimal way of making a change. Lighting a match on the street outside the studio ticked all our boxes. Photographing this moment, we returned to the studio and made life-size photograms using all the objects we kept in the studio. Making a picture out of tools, equipment and materials for the production of artworks placed a stronger emphasis on labour than we originally had in mind, and the persistence of the image of us lighting a match on the street continued to undermine the theme of the work.

Is the priority of the studio underlined by the use of this equipment or is it subsumed under and made subservient to the act of leaving the studio? The idea, rather, was to cross the line demarcating thought and action or art and life, typified by the doorway between the studio and the street, but also disclosed by paying attention to the means of production rather than to the product. We did not want to pass from one to the other (from object to process, from contemplation to action or from aesthetics to politics) but to hover between them or step back from them to see the bigger picture that contains them both. We wanted the studio to dream about the street and the street to dream about the studio. Above all, we wanted our artistic practice not only to incorporate process and to be located within the world adjacent to and beyond the place of production, but to convert these extensions of activity into the material body of the artwork itself.

We also made a series of paintings that had gone on a "day out". These were made in distinct phases. First, we painted a sheet of canvas as a flag—always tricolours but using earthy colours rather than the pure colours typical of national flags. Second, we took the flag for a walk, carrying it between us like a placard or banner. We got a friend to photograph the trip and then, finally, back in the studio, we painted the image of the flag on its "day out" back onto the flag itself. We painted in such a way that the flag, now functioning as a disruptive ground for the painting, could be partly seen through the image so, now, two images existed together on the same surface. The picture of an object in the world was superimposed on the object itself and therefore the flag motif appeared twice

at two different scales. The flag from the photographed action appeared both at a smaller scale in the image and at full lifelike scale through the picture, behind it, and existing in the here and now as a real thing.

The flag's day out was a deliberately awkward picture-object intended to sit on the axis of document, prop and action. At the same time, the flag works combined photography, performance, painting and sculpture in ways that meant it was unclear to which category or discipline they belonged. The separate media were detectable and real, but these were resolved best, we thought, through the narrative of what happened rather than through any putative properties of each discipline. The narrative of taking a canvas flag for a walk was the result of our reading of Art & Language's writing on "ofness" (Art & Language 1980), drawn from David Kaplan's concept of the difference between a name and a picture (Kaplan 1969) and Flint Schier's subsequent philosophy of what pictures are "of" (Schier 1986), which is to do with their genetic history rather than what they are perceived to resemble. This allowed us to think of picture making as necessarily a kind of performance or action that differed significantly from the examples typically extracted from Action Painting. Instead of dancing while painting, we might do something with a canvas in the way Beuys did things with rocks and trees and fat and felt. But rather than splitting the documentary photo off from the performance, we would collapse them into a single object that contained its own narrative of what it is "of" (Schier 1986).

With these ideas in mind, we looked into the possibility of making a new series of works that had a more complex history than a fake flag painted onto a sheet of canvas. We asked our friend John Wilkins to give us a few of his rejected paintings for us to use as the basis of a new set of works. We painted over his works in such a way that it looked like he had painted over our paintings. The defacement was real, but it looked fake and unconvincing and awkward. We painted a text "underneath" the figure and on top of the ground of his paintings. These were copied from photos of passages from an essay that we had crossed out. The work combined two real instances of rejection with two pictorial effacements that took place in an irreversible act of destruction made worse, in our minds, by the introduction of perspectival depth into a painting that dwelt in the shallow space of the printed page.

By making a work destroys a painting in its production, we linked the tradition of iconoclasm and the destruction of art to the philosophical inquiry into the practice of picturing and the relationship between the picture and the sequence of events that brings it about. In part, the idea was to imagine how the theory of picturing could be applied to the destruction of a painting, but it was also about thinking of the act of painting as doing something in the world. We thought it was worth risking the accusation of vandalism if that meant that the first thing people saw when they looked at the painting was a violent act and therefore, from our perspective, of emphasising the act of picturing rather than the pictorial character of the picture. These were the last works I made with Mark Hutchinson.

## 3. Scripts and Scores

My first solo works consisted of me performing to camera as various minimal monsters. I responded to the rise of "laddism" in yBa and Cool Britannia by reconnecting the "new lad" (a middle-class form of pastoralism (Crow 1993)) to older types of (working class) monstrous masculinity. Instead of dressing up and applying complex makeup, I would "transform" into a monster with the simplest of ready to hand things, in the tradition of street theatre. I was interested in how political street performers used props to represent multiple things—a stick used as a rifle, a crutch, a telescope, a wand and so on. Making one small change at a time, I put a mini torch down my underpants, stuck a calculator on my forehead, tied a toy horse's tail to the back of my trousers and wore a pair of cheap, plastic werewolf gloves. I took them using the self-timer on my camera to eliminate even the minimum "live" audience of a photographer or assistant. I used slide film in anticipation of displaying the work in the form of a slide carousel that displayed one mini monster after another in a kind of "flipbook" projected onto the wall of the gallery. I wanted to appear as

present in the space and imagined myself face-to-face with the viewer, sometimes facing them in a confrontational manner and sometimes with my back to them or ducking away from their gaze and so on.

Around the same time I made a video and a slide series (sufficient to fill a carousel) of young men making a gun shape with their fingers for a sequence that alternated between a shot of someone shooting the camera and a shot of someone acting in an amateurish way of being shot by the camera. Half the participants were artists and the other half were mates at the book warehouse where I worked. When Loaded presented itself as the magazine "for men who should know better" and John Currin was being admired for the "honesty" of his sexism, I drew on critical accounts of ultra-violent movies and subcultures (Hall and Jefferson 1975) related to hooliganism to think about the normalisation of nationalism, racism and masculinity (Rutherford 1997) during the rise of neo-fascism (Wank 1996). I used a childish and playful technique partly because of what I had read about the role of violence in the masculine rejection of boyhood (Jackson 1993), but also because it involved the kind of minimal transformation that I had used in my monster performances.

The formal structure of the images came from thinking about a certain crossover between theories of narrative and theories of metaphor. In particular, there is a curious similarity between, on the one hand, Frank Kermode's conception of narrative in *The Sense of an Ending*, which he illustrated with the false difference between the sounds "tick" and "tock" in the figure "tick-tock" that signifies the passing of time, and, on the other hand, Paul Ricoeur's assertion that "it always takes two ideas to make a metaphor" and William Empson's claim that metaphor asks the individual to hold two meanings in their head at the same time. I was interested in pairs of images, one tick and one tock, that both combined to make a mini totality that mimicked the video loop or GIF, and yet could sit easily within a longer sequence of iterable acts.

The shape of the works as a whole was an attempt to combine George Brecht's "event-score"—a simple repeatable scripted act—with the Minimalist structure of placing "one thing after another". Together, these established a formal "grid", more or less hidden underneath a "low" or everyday content that resonated more with Pop. The same logic was at play in "Dear Sarah", a series of daily faxes that I sent to Sarah Munro at the Collective Gallery in Edinburgh that requested the curator to visit a named individual taken from the gallery's mailing list for the purpose of discussing some issue of the day that I took note of during conversations with students.

Elements of this approach were evident in my contribution to the Bank exhibition, "Zombie Golf". After hearing their plans to produce life-size zombie figures to populate the exhibition, which I took to be a kind of provocation, I focused on one of the standard readings of zombies in cultural studies—as a trope for the less-than-human in the working class. I decided to fill the heads of these zombies with hope and fear. I made "thought bubbles" to hang around their heads like mobiles for infants. They were made out of photographs, two for each cloud. For one side, I wandered London with my camera and photographed small fragments of movie posters that expressed abstract, utopian hopes (for instance, "A Love Story Written in the Stars", "No One Stays at the Top Forever", "What Kind of Man Would Defy a King?" and "Beyond the Horizon Lies the Secret to a New Beginning"). For the other side, I went through books and magazines that I could get my hands on and photographed images of crowds screaming and shouting at sports and music events. The piece was called "The Road to Wembley" which was a deliberate downgrading of the title of Trotsky's essay "The Road to Socialism".

## 4. Pledges, Slogans and Manifestos

For the exhibition "Pledge" at Sparwasser HQ gallery in Berlin, I placed an ad in the local newspaper inviting people to nominate a daily walk (to work, school, the shops or whatever) and convert it into a political march by silently chanting a political slogan from history. People were asked to call the gallery and tell us the route and the slogan. During the period the exhibition a map of Berlin grew on the wall made up of lines of text that

followed Berlin's streets. Some areas were left completely blank, while others were densely represented by multiple "pledges" resulting in the short memorable texts overlapping and cutting across each other. At the time, I described this work as rooted in the power of words to transform, to trigger events and to activate the city. "Pledge" began as a written invitation (a performative speech act) which necessarily led to a linguistically transformed walk, linking daily activities to the protesters and marchers who fought for change and whose words changed the world.

In the cellar of the gallery, I performed in front of an audience. I sang in an amateur but sincere way after asking the audience not to *listen* to me (as we listen to singers) but to try to *hear* what I had produced. I had sent an email to everyone in my contact list, inviting everyone to write a song lyric. I explained in the invitation that I intended to sing all these lyrics from the heart as if I had written the words but also that I would sing the songs to tunes I have caught myself humming along to at home (mostly while doing the housework). Both works, therefore, focused on a private experience that opened up the individual to a mediated, lagged and estranged form of collectively. Also, the invitation to Berliners to pledge their daily walk to honour the protestors of the past is mirrored by my own pledge to everyone in my contact list to give voice to their words.

These were my last solo works before joining Mel Jordan and Andy Hewitt to form the Freee art collective. Our works consisted, mostly, of texts on walls and text-objects or photographs of texts and text-objects. We called the texts "slogans" and understood our use of language as "performative", both in the strict sense of "doing things with words" and in the more expansive sense of the discursive construction of solidarities and collective action. "Slogans" in our work were taken to be forms of language at the intersection of Conceptualism (specifically the anti-aesthetic significance of "Text Art") and protest. From our perspective, political slogans such as "Workers of the world unite!", "Votes for women" and "I <u>AM</u> a Man" are akin to "event-scores" insofar as they were intended to trigger specific repeatable actions that linked the individual to the collective.

Re-reading my notes on the Freee conception of the slogan and our practice of "sloganeering", I am struck by the transmission of the declarative tone from the slogan to the theory of the slogan:

> A slogan is a linguistic act that binds people as advocates for a partisan opinion or action. A short pithy phrase that obtains universal assent is not a slogan: a slogan divides the room. Rather than describing the political state of affairs, the slogan simultaneously diagnoses the problem with the existing society and prescribes a remedy. A slogan is not a description of the world but an intervention in it. Slogans are linguistic acts that aim to change the world. Slogans call for action. To interpret a slogan in the way literary critics interpret poetical phrases is to disable the slogan as a slogan. While aesthetic artworks derive part of their value from being unique, slogans live off repetition. All slogans want to be common property. Slogans want to be passed on. Slogans want to be reproduced and turn listeners into advocates. This is how slogans bind individuals into a social body. Individuals do not shout slogans; groups and crowds and mobs do.

The slogans in our work were intended as scripts for action related to a political history of declarative speech and the history of activist publishing, but where our use of language differed from both the history of Cagean instructions and the history of protest is that the "slogans" were introduced publics that were required to make a decision; we made works that asked members of the public to agree or disagree with us and with each other.

At first, we printed short texts on billboards. Our first body of work was a series of texts that were displayed as billboard sites on the street. The first was in Sheffield, the second in Venice and the third in London. The first one said: "The economic function of public art is to increase the value of private property." The second, written in English and Italian and two sides of a canal bridge, said: "The aesthetic function of art is to codify social distinctions as natural ones." And the third said: "The function of art for regeneration is to sex up the control of the underclasses." Eventually, we came to see these as lacking the embodied commitment and spatial particularity that are characteristic of chanting crowds.

We began, therefore, to produce the texts as objects and photographed ourselves holding or carrying the texts in specific places. The first was a floral funeral arrangement that said, "Protest is Beautiful"; the second, the phrase "Do not let the media have a monopoly on freedom of speech" printed across three T-shirts. Later, we also put texts on scarves, homemade badges, football shirts, pillowcases and balloons. We called the word-objects "props" because they were produced specifically to be used within a staged photograph, and we called the placing of props and people in specific locations, "real montage".

Freee also staged "spoken choirs". Along with participants, we would read aloud manifestos in public places. We wrote manifestos by taking a pencil (or a laptop) to an historical text, usually belonging to the entwined traditions of the avant-garde and political revolution. Sometimes, as Tristan Tzara advised, we chose the text according to its length; on other occasions, we selected the text according to the historical significance of the original, or in relation to the conditions of the invitation that prompted the writing of the manifesto. We wrote manifestos by editing and reworking *The Futurist Manifesto*, a Dada Manifesto and *The Communist Manifesto*, among others. The Freee manifesto *To Hell with Herbert Read* was written by changing the meaning of Herbert Read's essay "To Hell with Culture".

Each manifesto was prefaced with a set of instructions for readers:

*In order to participate you need to*

1. *print off the pdf (hard copies are also being distributed)*
2. *underline every sentence that you agree with*
3. *bring the manifesto to the event*
4. *read out those sections that you have underlined.*

This resulted in a collective reading that was very loud when everyone agreed or very quiet when only the three artists who wrote the manifesto agreed with the text. The performance of the text meant going public with one's private opinions that was registered in a strongly embodied form that often left people with a sore throat and a strange thrill of having got through to the end.

The last show we did together was "The Citizen Ship", a temporary public project that served as a publicly sited hub built for the purpose of opinion formation and vernacular publishing, commissioned by Milton Keynes Gallery in June 2017. When the Freee art collective dissolved, I decided to go solo again.

## 5. Proxy Publics

I wanted to build on the photographic mediation of social engagement in the work of Freee, which had often gone unnoticed or had not been fully appreciated, so I immediately liked the idea of underlining this by working alone but connected to the world photographically. At first, my model was "The Road to Wembley" and the slide works of monstrosity and violence that I had made in the 1990s. I made some videos of my finger tracing along the outlines of photos of robots in science fiction books while talking about how the publishing industry in the eighteenth century transformed the public into a spatially dispersed community of strangers.

I went in search of more books to use in videos of me tracing photographs and talking about technologies of public formation. In doing so I noticed that the act of shopping for books was an act of "real montage"—cutting things out of one context and pasting them into a new pattern in another context. I wanted the next step in the process to be more closely aligned with this activity of travelling and immersing myself in a crowd of people made out of books. I remembered my first photomontages at art school using stolen posters and started to cut up the images in the books and place them together in new combinations. I started with a double image, a metaphor or a tick-tock type narrative. My aim was not to make images (in the way that I imagine John Heartfield, Hannah Hoch, Peter Kennard and John Stezaker *make images*), but to make connections between places, people, things and events. My aim was to treat photomontage as an instance of "real montage".

The pioneers in the 1920s recognised that montage was not merely the editing of images together into a composite whole; it is evident also in the organisation of materials. Dziga Vertov rejected the idea that montage takes place only during the technical process of film editing. Rather, he argued, there is a "continuous process of editing" (Vertov 1984) from the selection of the theme (plucking it out of a heap of rival themes), the assemblage of research materials (newspaper clippings, books, photographs, maps, etc.), the plan of action ("the montage of your own observations or reports by informants and scouts"), the shooting plan ("selecting and sorting the human eye's observations") and so on. Alexander Rodchenko went further, saying montage "is a system of arranging things. Whether it is buttons and pockets on a suit, actors on a stage, soldiers in formation, furniture in a room, books on a shelf ... even a man's daily routine, divided into periods of work, rest and amusement" (Rodchenko 1978).

My starting point in making montages is buying second-hand books. I do this by visiting second hand bookshops, charity shops and so on. I start, in effect, on the street like the flaneur and the street photographer. I go to the street to find places where photographs are stored. All the second-hand books are waiting for a second (or third) chance of entering someone's life, someone's house, someone's bookshelf. They often have names or messages written on the inside cover. My starting point, then, presupposes an already existing flow. I start the work by joining the flow.

I visit local and regional second-hand bookshops to collect picture books that were once owned by members of the community. Just as a private book collection is a portrait of its owner, the montages that I make from visits to the bookshops of a particular region, taken as a whole, are portraits of the entire community. Instead of depicting what the people look like, the montages act as a map of what they look at and what they think about, what they value and what they collect. My intention is not to build my own collection from the dregs of other people's collections but to do something with the materials that I find.

The more I reflected on the spatial character of what I was doing—collecting books of photographs that had somehow made their way to me from every corner of the world—it seemed necessary for the montages to become more complex in order to house a greater variety of sources. Instead of working on the scale of the individual photograph, or the slightly larger scale of the photomontage that incorporated two or more images into a single image, I was now working on the scale of photography as a whole. Whereas Allan Sekula took on the role of a witness and therefore had to travel "to place the photographer and viewer at specific sites in the world", my montage practice is based on the act of compiling the activity of countless photographers from around the world in a combination of photographs that outstrips the capacity of any single observer.

The constructed image and the appropriated image of postmodern art shared a conviction in the reality of the photograph and the unreality of the photographed. My work understands the materiality of the photograph in a broader sense. Photographs are not merely images or merely pieces of paper; their materiality is also social, historical and economic. Photographs are a material element of the publishing industry in which photographers travel to certain places in order to make images that are subsequently selected, combined, captioned, printed, stored, shipped, sold and resold in a series of encounters, exchanges and transactions.

I travel, shop, collect, organise, archive and then, faced with boxes and bags on the floor of my studio, I choose a single book. It is the book, not the photograph, that is the unit of the photographic archive, and this is therefore the unit of the practice. Books are one of the principal ways in which photos pass into the world as things to be carried, held, moved and stored. Books of photographs are owned, gifted, cherished, thumbed and passed around. Books, therefore, are a part of the insistence that photographs are never immaterial and never merely signs or pictures. Whatever photos depict, they enact real and symbolic displacement, dispersal, distribution and dislocation. By taking fragments of individual photos out of books, therefore, I not only shuffle the archive but also rearrange the places, times, events, people and things that the history of photography has put into circulation.

The sequence of places along a path from the camera to the printer to the bookshelf are not legible in the image and so I have to chart the journeys they take, to reconstruct the passages from where each image was taken to all the places where the reproduced images appear before they enter the archive. I do not trace the outlines of photographed things, I run my fingers along the pages that inform the reader who took the image, where, when and why.

**Funding:** This research received no external funding.

**Data Availability Statement:** Not applicable.

**Conflicts of Interest:** The author declares no conflict of interest.

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
