# Peer review of "Picturing, Pledges and Other Scripted Acts: Performance (In) Art"

_arts_

Round 1
Reviewer 1 Report
Very interesting account of the performance behind the objects of contemporary art. I can see this argument carrying over into the inquiry regarding all artistic practices. I like the way that art making is embodied here.
Author Response
I appreciate the reviewer's support for the essay. There are no point-by-point comments so there is nothing else for me to do or say.
Reviewer 2 Report
You need to clarify the theories and the methods you're using. We don't really understand what your work adds to the scholarships. The piece seems to be a personal experience, not an academic piece.
Author Response
Let me to respond each of the 3 points.
- You need to clarify the theories and the methods you're using.
- We don't really understand what your work adds to the scholarships.
- The piece seems to be a personal experience, not an academic piece.
The first presupposes a model of research that is conflated with a certain type of writing and tone of voice. Current developments in research methodologies have challenged the longstanding confusion of research with certain conventions of presentation. Theory-fiction, practice led research and auto-ethnography are the most prominent examples of what is becoming a rich stream of alternative and experimental modes of conducting and presenting research. Your request for me to clarify my theories and methods is actually an attempt to deny these developments and call me back into the old style that this paper rejects. What I would say in response is not only that the paper adheres to contemporary research methods but also that it embodies its theories and methods rather than splitting theory and practice.
The second point has three answers because the paper contributes to scholarship in a number of ways. First, it applies knowledge from one field (performance art) to shed light on another (picture making in the expanded field of painting). Second, it discloses processes and ideas in the production of artworks that are not visible in the works themselves (this literally adds information that can be useful to scholars interested in contemporary art). Third, it identifies an error in established scholarship in art history which has restricted the discussion of performance in painting to dance-like processes of painterly technique (eg Jackson Pollock) and similar crossovers.
Third. Your accusation that the text is focused on personal experience without academic merit is unfair for two reasons. First, the paper is structured by numerous references to and discussions of important philosophical, sociological, art historical and other literature. Second, there is a long tradition of researchers using personal experience as a tool for academic research. Think of Richard Hoggart writing about his working class roots in The Uses of Literacy, or the work of numerous feminist academics who have used personal experience as a form of research based on the idea that the personal is political. In fact, it is no longer possible to draw a line between the personal and the academic in the way you imply. There is a difference between the research of an observer-participant and the research of a social scientist writing theories based on statistics, but this difference is not between what is of personal interest and what has academic value. Both are legitimate methods of conducting and presenting research to tte academic community and beyond.